# Benzimidazole Resistance in Cattle: The First Report of the Presence of F200Y Mutation in *Cooperia* in Ecuador

**DOI:** 10.3390/vetsci10060378

**Published:** 2023-05-28

**Authors:** Pamela Vinueza, Marlon Calispa, Luis Condolo, Paula Toalombo, Peter Geldhof

**Affiliations:** 1Escuela Superior Politécnica de Chimborazo, Facultad de Ciencias Pecuarias, Carrera de Medicina Veterinaria, Riobamba 060106, Ecuador; luis.condolo@espoch.edu.ec (L.C.); ptoalombo@espoch.edu.ec (P.T.); 2Laboratory of Parasitology, Faculty of Veterinary Medicine, Ghent University, Salisburylaan 133-B, 9820 Merelbeke, Belgium; peter.geldhof@ugent.be; 3UCLouvain, Earth al Life Institute ELIE, ELI Croix du Sud 2L/L7.05.05, 1348 Louvain-la-Neuve, Belgium; marlon.calispa@uclouvain.be

**Keywords:** anthelmintic, molecular, benzimidazole, *Cooperia* spp., Ecuador

## Abstract

**Simple Summary:**

Gastrointestinal nematode (GIN) infections constitute a serious threat to the cattle industry, and anthelmintics have been widely used to control them. As a result, resistance to broad-spectrum drugs is now a worldwide issue. In the present study, we evaluated the resistance status of GINs to fenbendazole (FBZ) in cattle of an Ecuadorian farm with a known history of broad-spectrum anthelmintic usage. We found that the GIN population was susceptible to FBZ and *Cooperia* spp. was the most prevalent genus before and after treatment. Although the reduction in egg count in *Cooperia* spp. was satisfactory (97.96%), we detected a phenylalanine-to-tyrosine substitution at codon 200 (F200Y) in the β-tubulin 1 gen in 43% of the pooled larva coproculture after treatment. The presence of F200Y suggests the existence of resistance in the early stages. Our findings highlight the need to implement alternative control strategies for parasitic infections besides the usage of broad-spectrum anthelmintics.

**Abstract:**

Anthelmintic resistance among GINs in cattle is a worldwide issue. Identifying the early signs of anthelmintic resistance (AR) is necessary to sustainably manage bovine parasitic infections. This study aimed to evaluate the resistance status of bovine parasitic nematodes against FBZ on a farm with a known history of broad-spectrum anthelmintic usage in Ecuador. FBZ efficacy was analyzed using a fecal egg count reduction test (FECR test) and β-tubulin 1 mutation identification in *Cooperia* spp., the dominant nematode parasite identified before and after treatment. According to the FECR test, the nematode population was susceptible to FBZ. After amplifying and cloning the β-tubulin 1 of *Cooperia* spp., an F200Y mutation was found in 43% of the pooled larva coproculture after treatment. This study reports, for the first time, the presence of F200Y resistance-conferring mutation in *Cooperia* spp. in Ecuador. Although the nematode population was phenotypically susceptible to FBZ, the presence of F200Y suggests the existence of resistance in the early stages. Our findings highlight the need to implement alternative control strategies for parasitic infections besides broad-spectrum anthelmintics.

## 1. Introduction

Gastrointestinal nematode infections are among the most prevalent livestock diseases in the world, and their impact on animal welfare and the economic sustainability of farms is enormous [1]. In response, control programs based almost exclusively on the usage of broad-spectrum anthelmintics have been implemented over the last 50 years [2]. Such programs and other factors have contributed to the development of AR in all major drug classes [3].

Ecuador is in northwestern South America and has a total of 25,637 million hectares of which 20% are destined for agricultural activity [4]. Of these, 57% of the area consists of natural and improved pastures, evidencing that livestock production is a significant agricultural sector of the country’s economy [4]. Cattle are the most important farming species, with 4 million heads, surpassing by a factor of four the next important livestock sector (pig production). Although together, sheep and goat enterprises represent less than 7% of the livestock market, all of them face the same challenges, such as low productivity or, more importantly, health issues [4,5].

Ecuador has several altitudinal thresholds, including lowlands with tropical climates [6], which are ideal for parasite development [7]. Moreover, open grazing is the primary husbandry system [4]; therefore, animals of all ages are potentially exposed to helminths all year round. Under this scenario, parasite control has been based on the preventive administration of broad-spectrum anthelmintic compounds [8,9] without paying attention to more sustainable integrated parasite control strategies. As a result, high selection pressure has been put on nematode populations, leading to widespread resistance in GIN cattle parasites to ivermectin (IVM) on various farms from the Coast, Sierra, and Amazon regions [10]. Although FBZ has shown high efficacy against GIN cattle parasites [10], resistance evolves, and previously susceptible worms can eventually develop into fully resistant nematodes [11]. Therefore, close monitoring of FBZ efficacy is needed to aid in the early detection of resistance and, ultimately, maintain its efficacy over the long term. The present study aims to evaluate the resistance status of bovine parasitic nematodes against BZ on a farm with a known history of this anthelmintic usage.

## 2. Materials and Methods

The farm where the study took place is in the province of Napo, El Chaco, and specializes in dairy cattle farming. On the farm, calves of different breeds (Normande, Jersey, and Holstein) are born all year round and weaned before the age of three months. The anthelmintic treatment regime on the farm was based on two drugs, triclabendazole and IVM, which were interchanged every four months.

### 2.1. Antiparasitic Treatment and Resistance Detection

A total of 15 calves ranging from 4 to 9 months were initially tested, but only 10 (females) were finally included. The decision was based on the minimum number of eggs per gram of feces (EPG) that calves should have in order to be included in the study (150 EPG). During the first day of treatment (D0), animals were weighed individually with an electronic scale and received a weight-based dose of anthelmintic (Panacur^®^ Merck, FBZ 10% oral administration, 5 mg kg^−1^).

Following BZ guidelines for resistance detection [1,12], fecal samples were obtained directly from the rectum on the day of the treatment (D0) and day 8 (D8). Fecal egg counts (FECs) were performed the same day using the Miniflotac technique [13] with a sensitivity of 5 EPG based on 45 mL of a saturated solution. To identify the nematode species, fecal samples from all calves were mixed, and a coproculture was performed on each sampling day following a well-known protocol previously described [10].

### 2.2. Statical Analysis

The treatment efficacy was calculated using the FECR test proposed in [14] and interpreted according to [15]. Consequently, FBZ resistance was considered present if the 95% upper (U95) and the 95% lower (L95) confidence intervals of the FECR percentage were below 95% and 90%, respectively. Similarly, susceptibility was inferred when the 95% upper (U95) and the 95% lower (L95) confidence intervals of the FECR percentage were above 95% and 90%, respectively. Efficacy against each genus was established based on the percentage of each nematode genus in the larva culture according to [16].

### 2.3. β-Tubulin 1 Mutation Identification

DNA was extracted from 3rd-stage larvae cultured at D8 using the PowerSoil^®^ DNA Isolation Kit (Mobio Laboratories Inc., Carlsbad, CA, USA). The β-tubulin 1 gene from *Cooperia* spp., the most prevalent nematode genera identified at D8, was amplified in a final volume of 25 μL using the forward and reverse primers described in [17]: (CoPCR167fw) 5′-TATGGGCACTTTGCTTATTTCA-3′ and (PCR198 + 200rev) 5′- CCGGACATYGTGACAGACACTAGG-3, respectively. Details on the reagents and parameters are listed in Table 1.

After gel electrophoresis, the tubulin gene was purified using a commercial kit (GENECLEAN^®^ II Kit, mpbio Laboratories Inc., Santa Ana, CA, USA), cloned into the pGEM^®^-T Easy Vector (Promega, Madison, WI, USA) and transformed into DH5α-competent cells. Finally, the plasmid products were sequenced bidirectionally. The sequences were blasted on NCBI and analyzed in Geneious [18] to identify BZ resistance-related codons in the β-tubulin 1 gene.

## 3. Results

Fecal samples of 10 female calves (4 to 9 months old) were included in the FECR test because they had 150 or more EPG of feces, as recommended in [1,12]. Egg counts per animal before and after treatment are presented in Table 2. The baseline levels of parasitic status before treatment were evaluated at D0; EPG ranged from 155 to 465. Egg reduction was checked at D8, where the maximum EPG count was 10. According to the FECR test and replicating our previous findings [10], FBZ was effective against the parasite population (FECR: 98.5% (U95 = 99; L95 = 98)).

An overview of the pre- and post-treatment larvae composition is presented in Table 3. *Cooperia* spp. was the most common species at D0 and D8, accounting for 54% and 73% of the parasite population, respectively. In addition, apart from *Ostertagia, Cooperia* spp. displayed lower susceptibility to FBZ when compared with the other genera. Nonetheless, larvae identified as *Haemonchus, Oesophagostomun, Ostertagia,* and *Bunostomun* were also recovered, yet at low proportions.

To ascertain BZ resistance-related mutations, the β-tubulin 1 gene from *Cooperia* spp., the most prevalent nematode genus identified before and after FBZ treatment, was amplified and cloned. Seven clones were finally sequenced and analyzed. Although the tubulin gene is strongly conserved among trichostrongylidos, differences in nucleotide sequence allow for the diagnosis of *Cooperia* at the species level [19]. According to the Local Alignment Search Tool [20], one of seven clones was identified as *C. oncophora*, while the remaining six belonged to *C. pectinata*. In addition, the PCR-based approach did not detect BZ resistance-associated alleles in codons F167Y and E198A. However, F200Y was detected in three of seven clones (43%) of *Cooperia*, specifically in *C. pectinata*.

## 4. Discussion

The BZ family, including FBZ, is composed of very effective drugs for treating nematode, tapeworm, and fluke infections, yet it exhibits little or no mammalian toxicity [21]. However, the fact that BZ lacks ectoparasite activity and has less convenient formulations (i.e., oral) has left most of the anthelmintic treatment dependent on its macrocyclic lactone (ML) counterparts [22]. According to the FECR test, FBZ was effective against the parasite population. In line with this observation, while widespread resistance in cattle to ML has been reported in North America, very few cases of BZ resistance have been described [23,24].

However, it is important to consider that phenotypic methodologies used to detect AR, such as the FECR test, possess their own drawbacks, as at the time AR is detected by this approach, at least 25% of the gastrointestinal nematodes carry the resistant alleles [1]. Under this scenario, resistance may be detected too late to implement mitigation strategies, such as the use of drug combinations or refugia management [2]. Compared to the FECR test, more sensitive techniques have been developed to detect BZ resistance in nematode parasites from ruminants. In this regard, three amino acid substitutions in the β-tubulin 1 gene have been associated with BZ resistance in several strongyle nematodes: phenylalanine-to-tyrosine substitution at codons 167 and 200 (F167Y, F200Y) and glutamic acid-to-alanine substitution in codon 198 (E198A) [25]. Here, we observed that F200Y was the only mutation detected at a frequency of 43%, specifically in *C. pectinata*. Across the world, this mutation is the most common aa substitution and has been shown to confer BZ resistance on its own in several natural field populations of parasitic nematodes, including *C. oncophora*, *C. punctata* [26,27], and *C. pectinata* [17]. However, to our knowledge, none of the known resistant mutations have been described in *Cooperia* spp. in Ecuador before.

Importantly, the genus *Cooperia* spp. has become the most prevalent cattle parasite observed in the United States, Australia, and Ecuador [10,28,29]. The reasons behind this phenomen may be related to the widespread use of anthelmintics and the subsequent selection pressure over the parasite population [24,30]. Under this scenario, the presence of the F200Y mutation in the surviving worm population and associated strong resistance to BZs in the present study suggest that a continuous use of these compounds can eventually lead to the replacement of the susceptible GIN population with a genetically resistant one. Although the genus *Cooperia* is, in general, acknowledged for causing mild symptomatology, infections with *C. pectinata* or *C. punctata* have shown to be much more severe, causing poor gain of weight, loss of appetite, diarrhea, or marked hypoproteinemia [31]. Consequently, failing to control resistant parasites will result in a profound impact on animal productivity in addition to the clinical effects [28].

As the development of AR is an evolutionary process, impossible to prevent when anthelmintics are used as part of a parasite control strategy [32], AR is redefining how parasite control should be practiced [11]. In this context, a deep knowledge of local GIN nematode epidemiology or refugia management would allow for applying a more holistic parasite control strategy [32], which may be a key factor in long-term profitable livestock production.

## 5. Conclusions

In conclusion, this study provides the first molecular evidence of BZ resistance in GIN parasites in Ecuador. As widespread resistance to ML compounds was previously detected, close monitoring of other anthelmintic compounds, including BZ, will be crucial to maintain drug efficacy over the long term. In addition, key information, such as nationwide prevalence and local epidemiological studies on parasites, will be necessary for evidence-based recommendations, ultimately leading to developing a more suitable control strategy.

## Figures and Tables

**Table 1 vetsci-10-00378-t001:** PCR reagents and parameters used to amplify the β-tubulin 1 gene from *Cooperia* spp.

Reagent	Parameter
dNTP: 0.25 mM	Initial denaturation: 98 °C for 30 s		
MgCl: 4 mM	Denaturation: 98 °C for 10 s	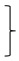	
Taq polymerase ^1^: 1 U	Annealing: 52 °C for 30 s	40 cycles
F ^2^-primer: 0.25 μM	Extension: 72° for 20 s	
R ^3^-primer: 0.25 μM	Final extension: 72° for 10 min		

^1^ GoTaq. ^2^ Forward. ^3^ Reverse.

**Table 2 vetsci-10-00378-t002:** Fecal egg counts in the feces of naturally infected calves before (D0 ^2^, N = 10) and after treatment (D8 ^3^) with FBZ 10% on a dairy cattle farm located in Napo Province, Ecuador.

Ear Tag	Age ^4^	EPG ^1^–D0	EPG ^1^–D8
895	4	465	5
12	5	215	5
13	5	705	10
10	6	290	0
11	4	275	0
14	5	485	5
SN	7	175	0
9	7	195	5
8	8	155	0

^1^ EPG: eggs per gram. D0 ^2^: day 0. D8 ^3^: day 8. ^4^ Age given in months.

**Table 3 vetsci-10-00378-t003:** Larvae identification pre- and post-treatment with FBZ 10% and efficacy of the anthelmintic treatment in a dairy cattle farm located in Napo Province, Ecuador. Data are presented in %.

Day	Coo. ^1^	Hae. ^2^	Oes. ^3^	Ost. ^4^	Bun. ^5^
Day 0	54	24	11	6	5
Day 8	73	6	2	19	0
Efficacy	97.96	99.62	99.73	95.23	100

^1^ *Cooperia*. ^2^ *Haemonchus*. ^3^ *Oesophagostomun*. ^4^ *Ostertagia*. ^5^ *Bunostomun*.

## Data Availability

The datasets utilized for this study are available from the corresponding author upon request.

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
