# Peer review of "Benzimidazole Resistance in Cattle: The First Report of the Presence of F200Y Mutation in Cooperia in Ecuador"

_vetsci, 2023, doi:10.3390/vetsci10060378_

Round 1
Reviewer 1 Report
General. The manuscript is rather short and with limited content, hence is should be resubmitted as communication.
Introduction. Please put the work in context. Add some more references regarding AR in Ecuador in catlle, sheep, goat parasites and the importance for health management of ruminants in the country.
M & M. 2.3. Please provide all the details of the PCR (not just the primers); the information must be presented in a table. As it is now, significant details of the work are missing.
Results. Please add a table with the results of detection of all the mutations, not just F200Y.
Discussion. This must be extended; as it is now, it is shallow and uninspiring. Also, it is unfortunate that the authors did not sample the animals at later days after the eight. This would have allowed more information. Anyway, the authors must justify this omission in the light of various relevant references.
Extensive editing of English language required.
Reviewer 2 Report
It deals with the present report of the manuscript that aims to evaluate the resistance of gastrointestinal nematodes in relation to the anthelmintic febendazole in a herd with a history of anthelmintic resistance (AR). The study is simple but presents very interesting results, mainly the possibility of early detection of AR through molecular techniques and has been associated with proliferation of resistant alleles in the parasite population.
It would be interesting to include in more depth practical aspects of how anthelmintic resistance can be delayed and even prevented. The use of parasitological diagnosis can identify animals that do not need to be treated on all occasions and these aspects could be discussed more emphatically.
Round 2
Reviewer 1 Report
The authors can add a brief passage with the clinical implications of their results, before acceptance of the revised manuscript.
Extensive editing of English language required.
